# Maternal body mass index and oxytocin in augmentation of labour in nulliparous women: a prospective observational study

Anna Ramö Isgren  , Preben Kjölhede , Sara Carlhäll, Marie Blomberg

Department of Obstetrics and Gynaecology, Department of Biomedical and Clinical Sciences, University Hospital, Linköping University, Linköping, Sweden

**Correspondence to**
Dr Marie Blomberg;
marie.blomberg@liu.se

## ABSTRACT

**Objective** To evaluate oxytocin use for augmentation of labour in relation to body mass index (BMI) on admission to the labour ward, focusing on cumulative oxytocin dose and maximum rate of oxytocin infusion during the first stage of labour.

**Design** Prospective observational study.

**Setting** Seven hospitals in Sweden.

**Participants** 1097 nulliparous women with singleton cephalic presentation pregnancy, ≥37 weeks of gestation, spontaneous onset of labour and treatment with oxytocin infusion for labour augmentation. The study population was classified into three BMI subgroups on admission to the labour ward: normal weight (18.5–24.9), overweight (25.0–29.9) and obese (≥30.0). The cumulative oxytocin dose was measured from the start of oxytocin infusion until the neonate was born.

**Primary outcome** Cumulative oxytocin dose.

**Secondary outcome** Maximum rate of oxytocin infusion during the active phase of first stage of labour.

**Results** The mean cumulative oxytocin dose increased in the BMI groups (normal weight 2278 mU, overweight 3108 mU and obese 4082 mU (p<0.0001)). However, when adjusted for the confounders (cervical dilatation when oxytocin infusion was started, fetal birth weight, epidural analgesia), the significant difference was no longer seen. The maximum oxytocin infusion rate during the first stage of labour differed significantly in the BMI groups when adjusted for the confounding factors individually but not when adjusted for all three factors simultaneously. In addition, the maximum oxytocin infusion rate was significantly higher in women with emergency caesarean section compared with women with vaginal delivery.

**Conclusions** Women with increasing BMI with augmentation of labour received a higher cumulative oxytocin dose and had a higher maximum oxytocin infusion rate during first stage of labour, however, when adjusted for relevant confounders, the difference was no longer seen. In the future, the guidelines for augmentation of labour with oxytocin infusion might be reconsidered and include modifications related to BMI.

## INTRODUCTION

Maternal obesity is associated with increased risks for obstetric complications.[1–4] In the first stage of labour, obese women have been reported to have a slower progression with increased risks for dystocia which, in turn, are risk factors for chorioamnionitis, emergency caesarean section (CS) and major postpartum bleeding.[4–14] With a higher incidence of dystocia, women with obesity have an increased need for treatment with oxytocin infusion for augmentation during labour. Previous studies have indicated that women with higher body mass index (BMI) receive larger oxytocin infusion doses and higher rates of oxytocin infusion in augmentation and induction of labour.[15–20] However, there is heterogeneity in the design of these studies and the results are not entirely consistent. Today, the guidelines for oxytocin administration in augmentation of labour differ worldwide and none of the guidelines adjust for maternal weight or BMI.[21–23] More research is needed to establish evidence-based clinical guidelines for the use of oxytocin during labour in relation to BMI.

This study evaluates oxytocin use for augmentation of labour in nulliparous women with spontaneous onset at term in relation to maternal BMI on admission to the labour ward. The primary aim was to determine the cumulative oxytocin dose in relation to the woman's BMI. Secondary aim was to analyse

## Strengths and limitations of this study

► This is a large, prospective, multicentre study including seven hospitals in Sweden.
► The study population was restricted to nulliparous women in full-term pregnancy with spontaneous onset of labour.
► Detailed background characteristics of the study population included pregnancy, obstetric and neonatal characteristics.
► The national Swedish guidelines for the definition of spontaneous start of labour was used which may limit the generalisability of the study.

the association between maternal BMI and maximum rate of oxytocin infusion during the active phase of the first stage of labour.

## METHODS

This is a prospective observational multicentre study conducted between April 2017 and February 2019 in all seven departments of obstetrics and gynaecology in the southeast health region of Sweden.

Inclusion criteria were nulliparous women with a singleton cephalic presentation pregnancy, ≥37 weeks of gestation with spontaneous onset of labour (ie, group one in the Ten Group Classification System)[24] and known admission BMI, who were treated with oxytocin infusion for labour augmentation and the cumulative oxytocin dose was measured.

All participating maternity wards adhered to the same national clinical guidelines concerning the definition of spontaneous onset of labour and for augmentation with oxytocin during active labour.[21 25] Likewise, the wards had identical protocols for oxytocin augmentation. The Swedish recommendation for the definition of active spontaneous onset of labour states that at least two out of three of the following criteria have to be fulfilled: spontaneous rupture of the membrane, regular painful contractions (2–3/10 min) and the cervix dilated 4 cm or effaced and dilated more than 1 cm. In addition to these criteria, the labour should progress within the following 2 hours.[25] The recommendation for initiation of augmentation with oxytocin during the active phase of the first stage of labour was defined as delayed progress of 3 hours from the expected 1 cm/hour cervical dilatation and during the second stage defined as no descent for 1 hour or active pushing for more than 30 min.[21] The guidelines for starting the oxytocin infusion also entail continuous cardiotocography (CTG), for registration of the fetal heart rate and monitoring contractions of the uterus.

Oxytocin was administered as an intravenous infusion through a peripheral venous catheter by an electronic infusion pump (B Braun Infusomat Space or Alaris GP Plus). A standard solution for infusion of oxytocin was prepared by adding 1 mL (containing 5 units of oxytocin) to 500 mL of 0.9% sodium chloride, reaching a concentration of 10 mU/mL. The infusion was routinely started at an infusion rate of 3.3 mU/min and the rate was increased every 20 min by 3.3 mU/min at a time until 4–5 contractions/10 min were achieved and/or normal progress of labour occurred according to the partograph, a graphic record of the progress of labour. Hyperstimulation was defined as >5 contractions/10 min.[21] The cumulative oxytocin dose was automatically registered by the infusion pump and was measured from the start of the oxytocin infusion until the neonate was born. The attending midwife registered the rate of oxytocin infusion manually in the partograph continuously during labour and took the reading from the pump after delivery for registration of the cumulative oxytocin dose.

Maternal height was reported or measured at the first antenatal visit and maternal weight was measured at admission to the labour ward, which enabled calculation of admission BMI (kg/m$^2$). The study population was classified into three subgroups according to the WHO BMI classification based on their BMI on admission to the labour ward: normal weight (BMI 18.5–24.9), overweight (BMI 25.0–29.9) and obese (BMI ≥30.0).

Demographic and clinical data comprising maternal, obstetric and neonatal characteristics were prospectively recorded in standardised medical records (Obstetrix, Cerner). The maternal characteristics consisted of age, smoking habit, diabetes mellitus, hypertension disorder (including pre-eclampsia) and maternal weight in the first trimester and at delivery. The obstetric characteristics included gestational length at delivery, epidural analgesia, mode of delivery, occurrence of obstetric anal sphincter injury and postpartum haemorrhage. In addition, the obstetric characteristics included cervical dilatation at the start of oxytocin augmentation subdivided into three categories:<6 cm, 6–10 cm and fully dilated,[26] the time from the start of active labour until the start of oxytocin augmentation, the time from the start of oxytocin augmentation until delivery, the cumulative oxytocin dose from the start of oxytocin infusion to delivery and the maximum oxytocin infusion rate during the first stage of active labour.

The neonatal characteristics comprised fetal birth weight, Apgar Score (<7 and<4 at 5 min) and umbilical arterial pH <7.0.

### Statistics

Sample size estimation was based on the cumulative oxytocin dose, given the assumptions that the minimal clinically important difference in cumulative oxytocin dose, between groups was 1200 mU and that the SD of the cumulative oxytocin dose was 3000 mU. Thus, in order to detect a statistically significant difference at a 5% level with 80% power, the sample size was estimated to be approximately 100 in each group.

Data are presented as mean and 1 SD or number and per cent. Comparisons of nominal data were conducted by means of Pearson $\chi^2$ tests. Continuous data were compared using one-way analysis of variance (ANOVA) tests for normally distributed data and Kruskal-Wallis ANOVA and Mann-Whitney U tests for non-normally distributed data in the univariate analyses. The outcome measures were evaluated in multivariate analyses by means of two-way ANOVA or analysis of covariance (ANCOVA), adjusting for the confounding factors cervical dilation at start of oxytocin infusion, categorised in <6 cm, 6–10 cm and fully dilated, fetal birth weight (as a continuous variable) and use of epidural analgesia (dichotomised in yes or no). In the multivariate analyses, the non-normally distributed continuous data variables underwent logarithmic transformation. Post hoc tests were performed by means of Tukey's honestly significant difference test. The level of significance was

**Table 1** Maternal, obstetric and neonatal characteristics in nulliparous women with singleton, term pregnancy with cephalic presentation and augmentation of labour with oxytocin infusion according to admission in labour body mass index (BMI)

| | BMI 18.5–24.9 (n=132) | BMI 25.0–29.9 (n=456) | BMI ≥30.0 (n=509) | P values |
|---|---|---|---|---|
| Maternal characteristics | | | | |
| Maternal age (years) | 28.1 (4.1) | 28.8 (4.4) | 28.2 (4.2) | 0.09* |
| Smoking during pregnancy | 5 (3.8) | 11 (2.4) | 19 (3.7) | 0.46† |
| Diabetes mellitus | 0 (0) | 4 (0.9) | 8 (1.6) | 0.25† |
| Hypertension disorder (including pre-eclampsia) | 0 (0) | 7 (1.5) | 24 (4.7) | <0.01† |
| Maternal weight in first trimester (kg) | 56.1 (6.1) | 62.4 (6.7) | 78.3 (14.4) | <0.0001* |
| Gestational weight gain (kg) | 11.1 (3.6) | 14.2 (4.4) | 16.9 (6.6) | <0.0001* |
| Obstetric characteristics | | | | |
| Gestational age at delivery (days) | 281.4 (7.8) | 281.5 (7.7) | 282.4 (7.5) | 0.13* |
| Time from active labour until start of oxytocin (hours) | 8.6 (4.0) (n=105) | 9.0 (5.3) (n=385) | 8.4 (4.6) (n=422) | 0.23‡ |
| Time from start of oxytocin until delivery (hours) | 3.0 (2.7) | 3.8 (3.4) | 5.0 (4.1) | <0.0001‡ |
| Cervical dilatation at start of oxytocin | | | | |
| <6 cm | 16 (12.1) | 87 (19.1) | 144 (28.3) | <0.0001§ |
| 6–10 cm | 34 (25.8) | 142 (31.1) | 207 (40.7) | |
| Fully dilated | 82 (62.1) | 227 (49.8) | 158 (31.0) | |
| Mode of delivery: | | | | |
| Normal vaginal delivery | 107 (81.1) | 379 (83.1) | 405 (79.6) | <0.01§ |
| Instrumental vaginal delivery | 22 (16.7) | 59 (12.9) | 61 (12.0) | |
| Caesarean section | 3 (2.3) | 18 (3.9) | 43 (8.4) | |
| Obstetric anal sphincter injury | 8 (6.1) | 22 (4.8) | 28 (5.5) | 0.82† |
| Postpartum haemorrhage (mL) | 456 (350) | 454 (378) | 488 (363) | <0.01‡ |
| Postpartum haemorrhage>1000 mL | 10 (7.6) | 31 (6.9) | 40 (7.9) | 0.83† |
| Epidural analgesia (no. of women) | 96 (72.7) | 355 (77.9) | 415 (81.5) | 0.07† |
| Neonatal characteristics | | | | |
| Fetal birth weight (g) | 3329 (351) | 3515 (420) | 3661 (467) | <0.0001* |
| Apgar Score<7 at 5 min (no. of infants) | 1 (0.8) | 7 (1.5) | 11 (2.2) | 0.50† |
| Apgar Score<4 at 5 min (no. of infants) | 0 (0) | 0 (0) | 1 (0.2) | 0.56† |
| Umbilical arterial pH<7.0 (no. of infants) | 0 (0) (n=110) | 7 (1.5) (n=393) | 7 (1.4) (n=442) | 0.37† |

Figures denote mean and 1 SD or number and per cent.
*One-way ANOVA.
†Pearson $\chi^2$ (df=2).
‡Kruskal-Wallis ANOVA.
§Pearson $\chi^2$ (df=4).
ANOVA, analysis of variance.

set at $p<0.05$ for two-tailed tests. Statistical analyses were performed using the software TIBCO Statistica, V.13.5 (TIBCO Software, Palo Alto, California USA).

**Patient and public involvement**

Patients or the public were not involved in the design, or conduct, or reporting or dissemination plans of our research.

**RESULTS**

The selection of the study population is presented in the flow chart (online supplemental figure S1). A total of 1097 women were included in the study. Maternal, obstetric and neonatal characteristics of the study population in relation to BMI group are presented in table 1.

As shown in table 2, the cumulative oxytocin dose increased significantly with increasing BMI group

**Table 2** Cumulative oxytocin dose and maximum rate of oxytocin during active phase of first stage of labour according to admission in labour body mass index (BMI)

| | BMI 18.5–24.9 | BMI 25.0–29.5 | BMI ≥30 | ANOVA/ANCOVA Between BMI groups, p value | | | | |
| | | | | Crude* | Adjusted † | Adjusted ‡ | Adjusted § | Adjusted ¶ |
| --- | --- | --- | --- | --- | --- | --- | --- | --- |
| | (n=132) | (n=456) | (n=509) | | | | | |
| Cumulative oxytocin dose (mU) | 2278 (2748) | 3108 (3839) | 4082 (4895) | <0.0001 | 0.42 | <0.001 | <0.0001 | 0.88 |
| Independent predictive factors: | | | | | | | | |
| Cervical dilatation | | | | | <0.0001 | | | <0.0001 |
| Fetal birth weight | | | | | | <0.0001 | | <0.0001 |
| Epidural analgesia | | | | | | | <0.0001 | <0.0001 |
| | (n=49) | (n=220) | (n=347) | | | | | |
| Maximum rate of oxytocin during active phase of first stage of labour (mU/min) | 12.7 (7.1) | 13.6 (9.1) | 15.5 (9.5) | <0.01 | <0.05 | <0.05 | 0.01 | 0.10 |
| Independent predictive factor: | | | | | | | | |
| Cervical dilatation | | | | | <0.0001 | | | <0.0001 |
| Fetal birth weight | | | | | | <0.01 | | <0.0001 |
| Epidural analgesia | | | | | | | <0.01 | <0.01 |

Figures denote mean and SD.
*Kruskal-Wallis ANOVA.
†Adjusted for cervical dilatation at start of oxytocin infusion.
‡Adjusted for fetal birth weight.
§Adjusted for epidural analgesia.
¶Adjusted simultaneously for cervical dilatation at start of oxytocin infusion, fetal birth weight and epidural analgesia.
ANCOVA, analysis of covariance; ANOVA, analysis of variance.

(normal weight, mean 2278 mU (2748 mU); overweight, mean 3108 mU (3839 mU) and obese, mean 4082 mU (4895 mU); Kruskal-Wallis ANOVA, p<0.0001). However, when adjusting for all three confounders, the significant difference was no longer seen between the three BMI groups (table 2).

Each of the three confounders were strong independent predictors of the cumulative oxytocin dose, but when adjusted simultaneously together, they equalised the differences in cumulative oxytocin dose completely between the BMI groups (figure 1).

The maximum rate of oxytocin infusion during the active phase of the first stage of labour differed significantly in the BMI groups in the univariate analyses (normal weight (n=49), mean 12.7 mU/min (7.1 mU/min), overweight (n=220), mean 13.6 mU/min (9.1 mU/min) and obese (n=347), mean 15.5 mU/min (9.5 mU/min); Kruskal-Wallis ANOVA, p<0.01). Individually, the confounding factors did not influence this, but when the analysis was conducted with all three confounding factors simultaneously, the significant difference between the BMI groups was no longer seen (table 2, figure 2).

When analysing the subgroup of women with overweight and obesity and known maximum rate of oxytocin infusion during the active phase of the first stage of

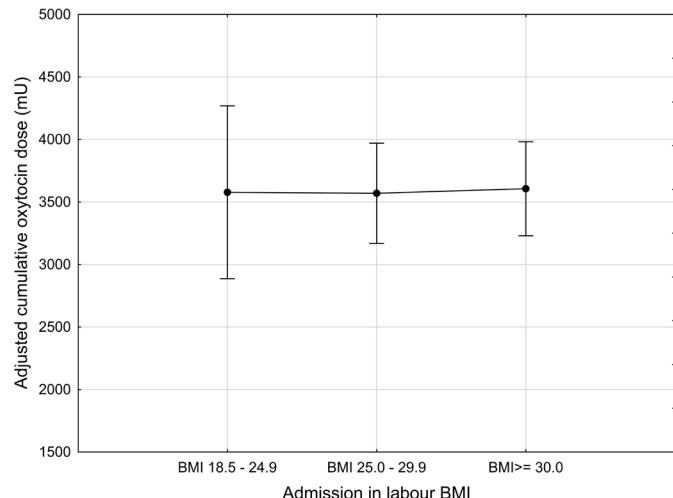

**Figure 1** Cumulative oxytocin dose in nulliparous women with singleton, term, pregnancy with cephalic presentation and augmentation of labour according to admission body mass index (BMI). Plots indicate mean and bars indicate 95% CI. Adjusted for cervical dilatation at start of oxytocin infusion, fetal birth weight and epidural analgesia.

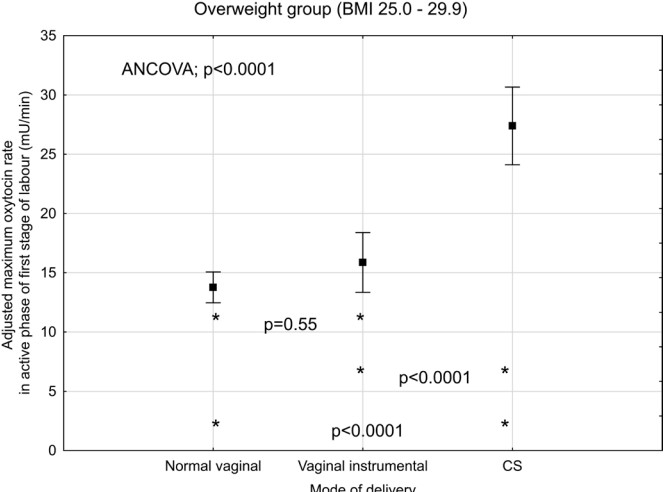

Figure 3 Maximum oxytocin rate in active phase of first stage of labour and mode of delivery in overweight, nulliparous women with singleton, term pregnancy with cephalic presentation and augmentation of labour. Plots indicate mean and bars indicate 95% CI. Adjusted for cervical dilatation at start of oxytocin infusion, fetal birth weight and epidural analgesia. ANCOVA analysis of covariance; BMI, body mass index; CS, caesarean section.

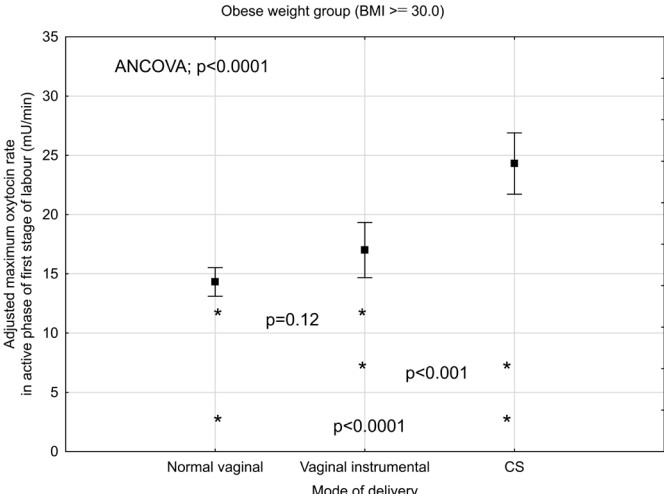

Figure 4 Maximum oxytocin rate in active phase of first stage of labour and mode of delivery in obese, nulliparous women with singleton, term pregnancy with cephalic presentation and augmentation of labour. Plots indicate mean and bars indicate 95% CI. Adjusted for cervical dilatation at start of oxytocin infusion, fetal birth weight and epidural analgesia. ANCOVA analysis of covariance; BMI, body mass index; CS, caesarean section.

labour, the maximum infusion rate was significantly associated with mode of delivery in both BMI groups when adjusted simultaneously for cervical dilation, fetal birth weight and epidural analgesia (figures 3 and 4).

The time from active labour until the start of oxytocin infusion did not differ between the BMI groups (Kruskal-Wallis ANOVA, p=0.23), (table 1), even when adjusted for fetal birth weight and epidural analgesia (ANCOVA, p=0.18). A significantly higher proportion of women with obesity ended with an emergency CS compared with

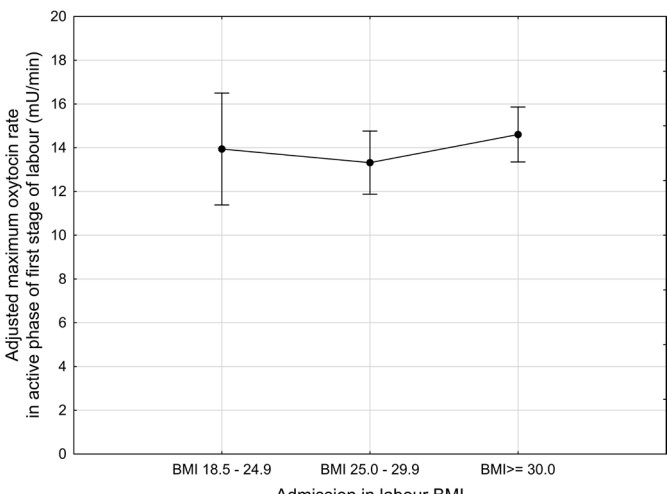

Figure 2 Maximum oxytocin infusion rate in first stage of labour in nulliparous women with singleton, term, pregnancy with cephalic presentation and augmentation of labour according to admission body mass index (BMI). Plots indicate mean and bars indicate 95% CI. Adjusted for cervical dilatation at start of oxytocin infusion, fetal birth weight and epidural analgesia.

normal weight and overweight women (8.4% vs 2.3% and 3.9%, respectively) (Pearson $\chi^2$ test; df=2, p<0.01), whereas no significant differences were seen between the BMI groups in women with a normal vaginal delivery or instrumental vaginal delivery.

The analysis of the women included in the study group versus those who were excluded mainly because of missing cumulative oxytocin dose showed similar proportions of characteristics, including indication for CS, in the two groups except for the rate of CS (5.8% in the study group vs 8.8% in the group of excluded women) and the occurrence of postpartum haemorrhage exceeding 1000 mL (7.4% vs 10.2%) (online supplemental table S1). Further analysis was conducted between the women who had CS with registration of cumulative oxytocin dose (n=64) and those women who had CS but oxytocin dose was not registered (n=106). Neither the degree of cervical dilation when oxytocin augmentation started differ nor did the time from the start of active labour to the start of infusion of oxytocin differ (data not shown). The time from the start of oxytocin augmentation to delivery was significantly shorter in those women whose cumulative oxytocin dose was registered (mean 4.1 hours (3.6 hours) vs 8.6 hours (5.1 hours)) (Mann-Whitney U test, p<0.01)).

## DISCUSSION
### Principal findings
The present study showed that nulliparous women at term with spontaneous onset and augmentation of labour in unadjusted analysis received a higher cumulative oxytocin dose and had a higher maximum oxytocin infusion rate at first stage of labour with increasing BMI on admission to the labour ward. After adjusting for cervical dilation at

start of oxytocin infusion, fetal birth weight and epidural analgesia, the significant difference was no longer seen. In addition, the maximum oxytocin infusion rate in first stage of labour was significantly higher in women with emergency CS compared with vaginal delivery.

## Strengths and limitations

The major strength of the study is the large number of participants restricted to nulliparous women with spontaneous onset of labour. The multicentre design made it possible to include women with different sociodemographic backgrounds, including both urban and rural areas. In addition, the labour wards included in the study had the same clinical guidelines for obstetric care.

The study has some limitations. Since there are no known cut-off values for BMI during pregnancy, we used the definition according to the WHO criteria. There are some limitations with the use of admission in labour BMI. Except increased fat mass, increased volume of extravascular fluid and blood and increased weight from maternal tissue from breast and uterus, admission BMI also includes the weight of the fetus, placenta and amniotic fluid. However, these limitations do not affect the purpose of this study, which was to find a relationship between BMI at admission to the labour ward and the cumulative oxytocin dose and maximum oxytocin infusion rate in the first stage of labour.

There might be a risk of selection bias since the cumulative oxytocin dose was not registered for all women. However, in order to determine the potential scope of selection bias, we compared the study group with the women that were excluded from the study. The data showed that the groups were similar except for the proportions of CS and postpartum haemorrhage, which were much higher in the excluded group. A plausible explanation for this discrepancy might be the time pressure that the staff were exposed to when a delivery was converted to an emergency CS or heavy postpartum bleeding occurred, and thus they forgot to record the cumulative oxytocin dose. All these aspects speak against selection bias.

Another limitation is our definition of the start of active labour. We used the national Swedish guidelines for the definition of spontaneous start of labour. There is no consensus worldwide either of the definition of spontaneous start of active labour or the definition of dystocia and this may limit the generalisability of the study. All participating sites in this study have well-established routines if hyperstimulation occurs (eg, reduce or stop the oxytocin infusion); unfortunately, we did not have the exact number of periods of hyperstimulation registered as the CTG traces were not scrutinised in the study.

## Interpretation in relation to other studies

This prospective study showed that women with increasing BMI received a higher cumulative dose of oxytocin compared with normal weight women; however, when adjusting for cervical dilation at start of oxytocin infusion, no significant difference between the BMI groups was seen. Soni et al[15] also studied women with spontaneous start of labour and treatment with oxytocin for arrest of dilation. They found that the total amount of oxytocin administrated was significantly higher in a group of women with admission BMI >35 (n=23) compared with women in three lower BMI groups. The BMI groups did not differ according to epidural or birth weight; however, the cervical dilatation at arrest differed significantly and were not adjusted for in the study.[15]

The present study showed that women with increasing BMI received a higher maximum infusion rate of oxytocin for augmentation in the first stage of labour even after adjustment for cervical dilation at start of oxytocin infusion. When additionally adjusting for fetal birth weight and epidural analgesia, the difference was no longer seen. The maximum oxytocin infusion rate in the first stage of labour in relation to maternal BMI has not been studied before in women with spontaneous start of labour. However, Hill et al[16] showed in induced labour that obese women (BMI>40) had a significantly higher mean maximum oxytocin infusion rate during first stage of labour compared with a group of lean women (BMI<28), matched for gestational age, fetal birth weight, cervical dilatation and fetal station at admission. Adams et al[19] studied oxytocin use in women with induced and spontaneous onset of labour and demonstrated that women with class III obesity had an increased risk for an oxytocin rate>20 mU/min, from admission until vaginal delivery, compared with lower BMI groups. However, in a subanalysis on spontaneous onset of labour, no significance was seen.[19]

The present study showed that women with obesity started oxytocin infusion at an earlier cervical stage compared with lower BMI groups. The reason for this can be discussed. Earlier studies have showed a longer first, but not second, stage of labour in obese women.[4 8 14] The time point when the oxytocin infusion starts equals when labour dystocia is diagnosed. Obese women received oxytocin infusion at an earlier cervical dilatation stage which could indicate an abnormal uterine contractility. Underlying pathophysiological mechanisms and ineffective uterine contractility among obese women have been studied. From biopsies, Zhang et al[6] showed that the myometrium from obese women undergoing elective CS at term contracted with less frequency and force compared with the myometrium from normal weight women. Other studies have indicated a different metabolic profile in obese women that could affect the contraction strength in the myometrium in an inhibitory way.[27 28]

## Clinical implications

There is evidence that oxytocin administration increases the risk for uterine hyperstimulation during labour, which can lead to fetal heart rate abnormalities. Furthermore, observational studies have shown an association between oxytocin use and increased risk for postpartum haemorrhage.[29] Therefore, clinicians must use oxytocin with

caution with respect to the potential risks and simultaneously be aware of the consequence of a CS due to dystocia and insufficient treatment with oxytocin in obese women. Our study showed that the maximum oxytocin infusion rate was significantly increased in overweight and obese women with emergency CS compared with vaginal deliveries. The reason for this is unclear but the explanation could be labour dystocia that did not respond to oxytocin treatment or fetal distress due to uterine hyperstimulation. Further research is needed to explore why women respond differently to oxytocin treatment. Perhaps, there is a different bodily distribution of administrated oxytocin in different BMI groups.

## CONCLUSION AND FUTURE DIRECTIONS

This large prospective study showed that women with increasing BMI received a higher cumulative oxytocin dose and higher maximum oxytocin infusion rate during the active phase of first stage of labour; however, when adjusting for cervical dilation at start of oxytocin infusion, fetal birth weight and use of epidural analgesia, the significant difference disappeared.

Oxytocin is one of the most commonly used drugs in the maternity wards and obesity is an increasing and major concern in obstetric healthcare. There are still knowledge gaps and further research is needed regarding the pathophysiological mechanisms in obese women affecting the dosage and duration of oxytocin in labour augmentation. In order to optimise treatment of labour dystocia, the patient care has to be individualised and the guidelines for augmentation of labour with oxytocin infusion might in the future be reconsidered and include modifications related to maternal BMI.

**Acknowledgements** The authors would like to thank the midwives and staff of the participating hospitals for collecting data.

**Contributors** All authors conceived the study idea, interpreted the data analysis, critically revised the manuscript and agree to take responsibility for the work. ARI acquired the data. PK and ARI performed the data analysis.

**Funding** This work was supported by the Medical Research Council of Southeast Sweden (FORSS) (grant no. FORSS-756621).Unrestricted grants were obtained from the County Council of Östergötland and Linköping University, Sweden. The funders were not involved in the development of the study from idea to project plan, implementation, analysis of data or writing of the manuscript.

**Competing interests** None declared.

**Patient consent for publication** Not required.

**Ethics approval** The regional ethical review board in Linköping, Sweden, approved the study (Dnr 2017/277-31). Need for informed consent has been waived for the study by ethics committee. All methods were performed in accordance with the relevant guidelines and regulations.

**Provenance and peer review** Not commissioned; externally peer reviewed.

**Data availability statement** Data are available upon reasonable request and according to Swedish legislation. Data statement section, technical appendix, statistical code and dataset are available from the authors.

**ORCID iDs**
Anna Ramö Isgren http://orcid.org/0000-0002-0253-1273
Preben Kjölhede http://orcid.org/0000-0001-5702-4116
Marie Blomberg http://orcid.org/0000-0003-4679-550X

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
