## [Reviewer comments · BMJ Open]

ARTICLE DETAILS

TITLE (PROVISIONAL)	Maternal body mass index and oxytocin in augmentation of labour in nulliparous women: a prospective observational study
AUTHORS	Ramö Isgren, Anna; Kjolhede, Preben; Carlhäll, Sara; Blomberg, Marie

VERSION 1 – REVIEW

REVIEWER	ANA ISABEL COBO-CUENCA UNIVERSIDAD DE CASTILLA LA MANCHA, SPAIN
REVIEW RETURNED	17-Oct-2020

GENERAL COMMENTS	Title: Maternal body mass index and oxytocin in augmentation of labour in nulliparous women: a prospective observational study The aim of this study is to know the differences between the use of oxytocin doses with nulliparous women depending on the BMI during of delivery. This study is a prospective observational study with 1,097 nulliparous women Congratulations to the authors, in my opinion it is a good study. The introduction is fine, but without updated references. I recommend updating the bibliography The methods section is very easy to read. I don't find in the manuscript the number of the ethical committee, I suggest the authors to add ethical aspects. Results Congrats on the flow chart. In the table 1, I suggest adding a column with the p-value of the comparison between all variables Discussion I suggest that the PRINCIPAL FINDING AND STRENGTH AND LIMITATION be removed and replaced the last paragraph of the discussion
--

REVIEWER	Constable, Laura Deakin University
REVIEW RETURNED	24-Oct-2020

GENERAL COMMENTS	General This is a prospective observational study that examined the cumulative oxytocin dose and maximum oxytocin rate for augmentation of labour among 3 different BMI groups in nulliparous women \geq presenting with spontaneous onset of labour. The primary study factor was BMI and the outcome measure was cumulative oxytocin dose.
---

	I think this is an important topic in current obstetrics practice, with a vast increase in clinical practice caring for overweight and obese women. It is interesting that oxytocin dosing guidelines are based on units and rate, with protocols referring to dose/rate, with no consideration of weight. (Many of our other protocols in medicine would all include a dose/kilogram for commonly used medications) The interplay between weight and BMI and uterine contractions/response to oxytocin is still a topic not well understood (as discussed in discussion section) I think overall this is a very well designed study with sound statistical analysis I think it's an important topic and that will help guide considerations of oxytocin guidelines and considerations in current clinical care. Overall, I think this is very well designed study which is very clinically relevant. I would strongly support publication with some minor changes as outlined below. As usual, the authors should feel free to either make changes the suggested and can disagree, as the following comments are only my opinion. General comments: What was the consideration in terms of rupture of membranes and did you collect data on who had spontaneous rupture or artificial rupture of membranes? I note in your definition of spont labour that they needed 2/3 criteria, so may have undergone ARM . Did you measure time to reach rate of maximum oxytocin rate? In your study design, you identified 3 pre-selected confounding factors (cervical dilatation, fetal birth weight and epidural). What was this based on (previous studies)?Are you there any other potential confounding factors that could be contributing? Did you look at underlying indication for the augmentation? And whether that could have influenced how effective the oxytocin was or impacted on likelihood on have emergency C-section. I do agree that while all centres used Swedish guidelines for definition of spontaneous labour and for the augmentation oxytocin protocol, it does limit the generalisability as different countries use different protocols for dosing etc. I think it is important to consider the risks uterine hyperstimulation, uterine rupture etc when weighing up risks vs benefits of trialling higher doses. Abstract Line 36 - 'three admission in labour BMI subgroups' could be said a bit clearer, perhaps 'three BMI subgroups on admission to labour ward' Line 46 - 'Together, but not individually, the confounding factors eliminated statistical significance.' I don't quite understand what you mean - when adjusted for cervical dilatation only, the maximum infusion rate was statistifcally significant between BMI groups, but not when adjusted for all 3 confounding factors? Line 57 - can you clarify whether 6 or 7 hospitals Introduction Good introduction of key issue and why relevant to current practice Good to note that no guidelines worldwide incorporate or consider maternal weight in oxytocin dosing Line 80 - again the wording here is hard to follow 'in relation to maternal admission at labour BMI'
--	--

	Methods Line 92 - can you provide more clarity on what the definition of spont labour according to The Ten Group classification? I had finding it in reference 22 abstract, talking about rates of casarean section Good clarity on Swedish definition of spont labour Was there any consideration or collection of data on women requiring artificial ruptre membranes? Line 105 - 'guidelines for start of oxytocin infusion' grammar - '?starting oxytocin infusion' Again, good to mention that there are different guidelines in different countries - rates, interval to increase and desired targets As above, consideration of underlying indication for augmentation? Line 112 - long sentence, consider dividing into two sentences Description of cumulative oxytocin dose and rate of infusion - good BMI definition - good Were there any underweight women BMI <18 and if so, how were they treated? Diabetes mellitus - did this includes gestational diabetes or only pre-existing? Any insight into time at which maximum infusion rate was reached? Statistics Sample size -assumptions that the minimal clinically important difference in cumulative oxytocin dose, between groups was 1200 mU - what is this assumption based on? Pre-selected confounding variables - cervical dilation, epidural, fetal birth weight - what was this based on? Did you consider looking at other variables you collected data on to see if significantly associated with cumulative dose in univariate analysis? Results Flow chart - clear representation. Again, any BMI <18.0? Table 1 - shows maternal, obstetric and neonatal characteristics across 3 BMI groups. In other studies, Table 1 has often shown characteristics dichotomised by primary outcome. This would be has as you have continous variable outcome and yours is prospective study, so makes sense to reader to present the characteristics across BMI groups (to show no large significant differences between groups) Table 2 - shows crude and adjusted cumulative oxytocin dose by BMI group. Main results: Cumulative oxytocin dose - increased with BMI group but when adjusted for confounders, no significant difference Max rate - increased with BMI group, but when adjusted for confounders, no significant difference You could maybe avoid using word significant difference 'disappeared,' perhaps 'no longer present.' Discussion Even though after adjustment for your confounding factors, no longer statistical significant, I think your main result is relevant to know in clinical practice and for further research in this area.
--	--

	I think your statistically significant results - that in overweight and obese groups, there was a much higher maximum rate of infusion and a much higher rate of caesarean is really important. Do you think this is because as per augmentation protocol oxytocin rate continued to increase, and then when this failed, progress to emergency caesarean, or was there some effects of the higher infusion rate on fetal HR, uterine hyperstimulation etc? Line 286 - again, could reword 'admission in labour BMI' I think this is really important point from your study - 'Our study showed that the maximum oxytocin infusion rate was significantly increased in overweight and obese women with emergency CS compared with vaginal deliveries. The reason for this is unclear but the explanation could be labour dystocia that did not respond to oxytocin treatment, or fetal distress due to uterine hyperstimulation. Further research is needed to explore why women respond differently to oxytocin treatment.'
--	---

VERSION 1 – AUTHOR RESPONSE

Point-by-point response to reviewers comments

Reviewer: 1

Comments to the Author

Title: Maternal body mass index and oxytocin in augmentation of labour in nulliparous women: a prospective observational study

The aim of this study is to know the differences between the use of oxytocin doses with nulliparous women depending on the BMI during of delivery.

This study is a prospective observational study with 1,097 nulliparous women

Congratulations to the authors, in my opinion it is a good study.

- Thank you for the positive feedback.

The introduction is fine, but without updated references. I recommend updating the bibliography

- We have now updated the bibliography by adding two new references.

The methods section is very easy to read.

I don't find in the manuscript the number of the ethical committee, I suggest the authors to add ethical aspects.

- Ethical aspects have been added under the heading "Details of Ethical Approval"

Results

Congrats on the flow chart.

- Thank you

In the table 1, I suggest adding a column with the p-value of the comparison between all variables

- We have added the requested p-values in Table 1

Discussion

I suggest that the PRINCIPAL FINDING AND STRENGTH AND LIMITATION be removed and replaced the last paragraph of the discussion

- The sections PRINCIPAL FINDING and STRENGTH AND LIMITATION were placed according to the BMJ Open guidelines to the authors.

Reviewer: 2

Comments to the Author

General

This is a prospective observational study that examined the cumulative oxytocin dose and maximum oxytocin rate for augmentation of labour among 3 different BMI groups in nulliparous women \geq presenting with spontaneous onset of labour. The primary study factor was BMI and the outcome measure was cumulative oxytocin dose.

I think this is an important topic in current obstetrics practice, with a vast increase in clinical practice caring for overweight and obese women.

It is interesting that oxytocin dosing guidelines are based on units and rate, with protocols referring to dose/rate, with no consideration of weight. (Many of our other protocols in medicine would all include a dose/kilogram for commonly used medications)

The interplay between weight and BMI and uterine contractions/response to oxytocin is still a topic not well understood (as discussed in discussion section)

I think overall this is a very well designed study with sound statistical analysis

I think it's an important topic and that will help guide considerations of oxytocin guidelines and considerations in current clinical care.

Overall, I think this is very well designed study which is very clinically relevant. I would strongly support publication with some minor changes as outlined below.

As usual, the authors should feel free to either make changes the suggested and can disagree, as the following comments are only my opinion.

- We appreciate these thoughtful comments and positive feedback.

General comments:

What was the consideration in terms of rupture of membranes and did you collect data on who had spontaneous rupture or artificial rupture of membranes? I note in your definition of spont labour that they needed 2/3 criteria, so may have undergone ARM .

- Unfortunately we did not extract data regarding spontaneous rupture or artificial rupture of membranes.

Did you measure time to reach rate of maximum oxytocin rate?

- No, we did not register the time to reach maximum oxytocin rate.

In your study design, you identified 3 pre-selected confounding factors (cervical dilatation, fetal birth weight and epidural). What was this based on (previous studies)? Are there any other potential confounding factors that could be contributing?

- Thank you for this wise reflection. We carefully considered the selection of potential confounders. The use of these in the multivariable models was based on the literature. Other factors or covariates may be confounders, such as age and gestational age, but since these covariates were equal in the BMI groups, we intentionally abstained from adding them to our confounders.

Did you look at underlying indication for the augmentation? And whether that could have influenced how effective the oxytocin was or impacted on likelihood on have emergency C-section.

- The indication for the augmentation with oxytocin during the active phase of the first stage of labour was defined as delayed progress of three hours from the expected 1 cm/hour cervical dilatation and during the second stage defined as no descent for one hour, or active pushing for more than 30 minutes. In active labour, it is often difficult to specify underlying factors that contributes to the dystocia. In some cases the reason for the dystocia will be obvious at the time of the Caesarean section or after birth (eg. asynclitism, narrow pelvis). From our clinical point there is usually more than one factor that contributes to the lack of progress. The question whether the underlying indication for the augmentation could have influenced the effectiveness of oxytocin is very interesting and deserves attention, our study was not aimed to elucidate this issue.

I do agree that while all centres used Swedish guidelines for definition of spontaneous labour and for the augmentation oxytocin protocol, it does limit the generalisability as different countries use different protocols for dosing etc.

I think it is important to consider the risks uterine hyperstimulation, uterine rupture etc when weighing up risks vs benefits of trialling higher doses.

- The authors totally agree, both generalisability and safety are pointed out in the discussion section.

Abstract

Line 36 - 'three admission in labour BMI subgroups' could be said a bit clearer, perhaps 'three BMI subgroups on admission to labour ward'

- We appreciate the advice and accordingly we have rephrased it in the manuscript.

Line 46 - 'Together, but not individually, the confounding factors eliminated statistical significance.' I don't quite understand what you mean - when adjusted for cervical dilatation only, the maximum infusion rate was statistically significant between BMI groups, but not when adjusted for all 3 confounding factors?

- Yes, that is correct. We have rephrased the sentence to make it, hopefully, unambiguous.

Line 57 - can you clarify whether 6 or 7 hospitals

- Sorry for this lapse. The study was conducted at seven hospitals. The number has been corrected in the manuscript.

Introduction

Good introduction of key issue and why relevant to current practice

Good to note that no guidelines worldwide incorporate or consider maternal weight in oxytocin dosing

Line 80 - again the wording here is hard to follow 'in relation to maternal admission at labour BMI'

- See previous response to the same comment from Reviewer #2 above.

Methods

Line 92 - can you provide more clarity on what the definition of spont labour according to The Ten Group classification? I had finding it in reference 22 abstract, talking about rates of caesarean section

- The Ten Group classification system (TGCS) is a prospective standardized structure of clinically relevant groups of women. Spontaneous start of labour is not defined according to the TGCS.

Good clarity on Swedish definition of spont labour

Was there any consideration or collection of data on women requiring artificial rupture membranes?

- See previous response to the same comment from Reviewer #2 above.

Line 105 - 'guidelines for start of oxytocin infusion' grammar - '?starting oxytocin infusion'

- Has been changed accordingly.

Again, good to mention that there are different guidelines in different countries - rates, interval to increase and desired targets

As above, consideration of underlying indication for augmentation?

- See previous response to the same comment from Reviewer #2 above.

Line 112 - long sentence, consider dividing into two sentences

- The sentence has been revised.

Description of cumulative oxytocin dose and rate of infusion - good

BMI definition - good

Were there any underweight women BMI <18 and if so, how were they treated?

- None of the women had BMI < 18.5

Diabetes mellitus - did this include gestational diabetes or only pre-existing?

- Diabetes mellitus included both gestational and pre-existing diabetes mellitus

Any insight into time at which maximum infusion rate was reached?

- See previous response to the same comment from Reviewer #2 above.

Statistics

Sample size - assumptions that the minimal clinically important difference in cumulative oxytocin dose, between groups was 1200 mU - what is this assumption based on?

- The assumption that the minimal clinically important difference between groups of 1200 mU was based on empiric clinical judgements from the senior obstetricians in the research group.

Pre-selected confounding variables - cervical dilation, epidural, fetal birth weight - what was this based on?

- See previous response to the same comment from Reviewer #2 above.

Did you consider looking at other variables you collected data on to see if significantly associated with cumulative dose in univariate analysis?

- See previous response to the same comment from Reviewer #2 above.

Results

Flow chart - clear representation. Again, any BMI <18.0?

- None of the women had BMI < 18.5

Table 1 - shows maternal, obstetric and neonatal characteristics across 3 BMI groups. In other studies, Table 1 has often shown characteristics dichotomised by primary outcome. This would be has as you have continuous variable outcome and yours is prospective study, so makes sense to reader to present the characteristics across BMI groups (to show no large significant differences between groups)

- Agree

Table 2 - shows crude and adjusted cumulative oxytocin dose by BMI group.

Main results:

Cumulative oxytocin dose - increased with BMI group but when adjusted for confounders, no significant difference

Max rate - increased with BMI group, but when adjusted for confounders, no significant difference

You could maybe avoid using word significant difference 'disappeared,' perhaps 'no longer present.'

- Thank you for this suggestion; we have accordingly revised the text as suggested

Discussion

Even though after adjustment for your confounding factors, no longer statistical significant, I think your main result is relevant to know in clinical practice and for further research in this area.

I think your statistically significant results - that in overweight and obese groups, there was a much higher maximum rate of infusion and a much higher rate of caesarean is really important. Do you think this is because as per augmentation protocol oxytocin rate continued to increase, and then when this failed, progress to emergency caesarean, or was there some effects of the higher infusion rate on fetal HR, uterine hyperstimulation etc?

- Thank you for an interesting obstetric reflection. To study indications for emergency CS are, in general, complicated. You have to create a hierarchy of indications, as the majority of CSs have more than one indication. The impact of the maximum oxytocin rate on uterine contractility and fetal wellbeing is highly interesting and warrants future studies

Line 286 - again, could reword 'admission in labour BMI'

- The sentence has been revised for better clarity.

I think this is really important point from your study - 'Our study showed that the maximum oxytocin infusion rate was significantly increased in overweight and obese women with emergency CS compared with vaginal deliveries. The reason for this is unclear but the explanation could be labour dystocia that did not respond to oxytocin treatment, or fetal distress due to uterine hyperstimulation. Further research is needed to explore why women respond differently to oxytocin treatment.'

VERSION 2 – REVIEW

REVIEWER	Laura Constable Royal Children's Hospital, Melbourne
REVIEW RETURNED	17-Jan-2021
GENERAL COMMENTS	Congratulations to the authors on this revision. They have revised the areas identified in the previous review and would fully support publication.